# Feasibility of SMS booster for alcohol reduction in injury patients in Tanzania

Catherine A. Staton[1,2]*, Mary Catherine Minnig[2], Ashley J. Phillips[1], Arthi S. Kozhumam[2], Msafiri Pesambili[3], Brian Suffoletto[4], Blandina T. Mmbaga[2,3,5,6], Kennedy Ngowi[5], Joao Ricardo Nickenig Vissoci[1,2]

1 Department of Emergency Medicine, Duke University School of Medicine, Durham, North Carolina, United States of America, 2 Duke Global Health Institute, Duke University, Durham, North Carolina, United States of America, 3 Kilimanjaro Christian Medical Center, Moshi, Tanzania, 4 Department of Emergency Medicine, Stanford University, California, United States of America, 5 Kilimanjaro Clinical Research Institute, Moshi, Tanzania, 6 Kilimanjaro Christian Medical University College, Moshi, Tanzania

* catherine.staton@duke.edu

**Data Availability Statement:** The authors do not have permission to share the data widely according to our regulatory agencies. As such, we can make the data available upon request to a third party

## Abstract

Alcohol use is associated with 3 million annual deaths globally. Harmful alcohol use, which is associated with a high burden of disease in low- and middle-income countries (LMICs), often increases the probability of traumatic injury. Treatments for harmful alcohol use in LMICs, such as Tanzania, lack trained personnel and adequate infrastructure. The aim of this study was to assess the feasibility of using SMS boosters to augment a hospital based brief negotiational intervention (BNI) in this low resourced setting. We conducted a three stage, four arm feasibility trial of a culturally adapted BNI for injury patients with harmful and hazardous drinking admitted to Kilimanjaro Christian Medical Centre (KCMC) in Moshi, Tanzania. Post hospital discharge, two of the four arms included patients receiving either a standard or personalized short message service (SMS) booster to enhance and or perpetuate the effect of the in-hospital BNI. Text messages were sent weekly throughout a 3-month follow-up period. SMS feasibility was assessed according to the TIDier checklist evaluating what, when, how much, tailoring processes, modifications and how well (intervention fidelity). Data was collected with SMS logs and short answer surveys to participants. A total of 41 study participants were assigned to each receive 12 SMS over a three-month period; 38 received messages correctly, 3 did not receive intended messages, and 1 received a message who was not intended to. Of the 258 attempted texts, 73% were successfully sent through the messaging system. Of the messages that failed delivery, the majority were not able to be sent due to participants traveling out of cellular service range or turning off their phones. Participants interviewed in both booster arms reported that messages were appropriate, and that they would appreciate the continuation of such reminders. At 6-month follow-up, 100% (n = 11) of participants interviewed believed that the boosters had a positive impact on their behavior, with 90% reporting a large impact. This study demonstrated feasibility and acceptability of the integration of SMS mobile health technology to supplement this type of nurse-led BNI. SMS booster is a practical tool that can potentially prolong the impact of a brief hospital based intervention to enact behavioral change in injury patients with AUD.

Gwamaka William, gwamakawilliam14@gmail.com, at the agencies National Institute of Medical Research in Tanzania and the Kilimanjaro Christian Medical Center Ethics Committee.

**Funding:** This project was conducted with funding from the National Institute of Health Fogarty International Center (K01-TW010000-01A1, PI CAS). We report no conflicts of interest. The funders had no role in study design, data collection and analysis, decision to publish, or preparation of the manuscript.

**Competing interests:** The authors have declared that no competing interests exist.

## Introduction

Alcohol use contributes to health conditions that cause more than 60% of deaths. In addition to being linked with 60 acute and chronic diseases, alcohol use is associated with 3 million deaths annually [1, 2]. Low- and middle-income countries (LMICs) account for the highest burden of alcohol use and the related morbidity globally [3]. Africa disproportionately holds the highest burden of disease attributable to alcohol use [1, 4]. In sub-Saharan Africa, unintentional injuries account for the majority of alcohol-related deaths and disability adjusted life years (DALYs) lost in this region [5]. Previous studies conducted in Tanzania determined a high prevalence of harmful and alcohol use when compared to other sub-Saharan African countries. Harmful alcohol use in Tanzania has also been shown to be linked with lower socioeconomic status, unemployment, and male sex [6, 7]. Rates of alcohol consumption are higher in the Kilimanjaro region of Tanzania than in other Northern regions of the country likely as a result of differences in reporting measures, socioeconomic status, and other cultural practices [8, 9].

In high-income countries (HICs), brief negotiational interventions (BNIs) administered in emergency department settings have been widely studied and have been shown to reduce hazardous alcohol use and its related harms [10–13]. Meanwhile, low- and middle-income countries (LMICs) generally lack locally-derived evidence to identify effective and cost-efficient interventions; but a BNI and motivational interviewing techniques are among the most commonly evaluated and effective interventions [14]. Studies examining the most effective strategies for introducing BNIs in low-resource settings show mixed results [15, 16].

Internet-, text-, and smartphone-based programs, both stand-alone and in combination with others, have been used as interventions to prevent and mitigate alcohol and substance use [17]. Though short message service (SMS)-based interventions have been implemented in Sub-Saharan Africa [17–19], the majority have occurred in upper-middle and high income countries [20–24]. SMS text and booster interventions offer a low-cost, and highly available intervention for those using alcohol [24]. In Tanzania specifically, SMS interventions have been implemented to improve helmet wearing [25], cervical cancer screening awareness [26], and community health worker performance [27], however none as of yet have focused on alcohol use behavior.

This study tests the feasibility of SMS boosters which augmented the "Punguza Pombe Kwa Afya Yako" (PPKAY)/ "Reduce Alcohol for Your Health" intervention, which is a brief negotiational intervention created and piloted at the Kilimanjaro Christian Medical Centre (KCMC) in Moshi, Tanzania. PPKAY was designed and adapted using the American College of Emergency Physicians/ NIAAA-recommended standards [28]. The PPKAY was adapted to Swahili and the Tanzanian environment with help from a trained local research team with methods further delineated elsewhere [29]. A Pragmatic Randomized Adaptive Clinical Trial (PRACT) was designed to evaluate 1) whether the PPKAY would lead to a reduction in alcohol related harms at 3-month follow-up when compared to the usual standard of care; 2) whether a standard SMS booster would improve the effectiveness of the PPKAY intervention; and 3) whether a personalized SMS booster would be more effective than PPKAY alone or PPKAY with standard booster. This manuscript aims to describe enrollment, retention, process measures, and acceptability of the SMS booster portion of the intervention. Our feasibility assessment of the SMS booster integration was performed using a modified Template for Intervention Description and Replication (TIDieR) Checklist [30].

## Methods

### Ethical statement

This study was approved by the Duke University Medical Center Institutional Review Board (Pro00062061), Kilimanjaro Christian Medical Center Ethics Committee and the Tanzanian National Institute of Medical Research.

## Setting

This study took place at the Kilimanjaro Christian Medical Centre (KCMC), in Moshi, Tanzania. KCMC serves the Kilimanjaro Region of Northern Tanzania as a major referral center, admitting nearly 2,000 patients annually in the emergency department (ED) alone [31]. About 30% of adult injury patients who present to KCMC have a positive alcohol by breathalyzer on arrival, and prior work has shown there is limited patient knowledge about treatment options in the region [9, 31]. In the Kilimanjaro region, successful mobile health technology-based interventions have taken place providing a basis for this study [32–36].

## Participants

Adult patients ($\geq$ 18 years of age) who were seeking care at the KCMC ED for an acute (<24 hours) injury were identified through our pre-screening process. Of those who were pre-screened, those who were not found to be clinically intoxicated, as determined by the treating physician, and who were able to provide consent were offered informed consent to participate in this study. Individuals who were intoxicated or too ill to provide consent during initial evaluation were reassessed for ability to consent within the first 24 hours of hospital arrival. Ability to provide consent was determined through physical examination by the treating physician. Written informed consent was collected at this time for inclusion in the feasibility trial and to test for alcohol. To be included in this feasibility trial, patients either disclosed alcohol use prior to injury, received a $\geq$8 on our Swahili validated Alcohol Use Disorders Identification Test (AUDIT) [37], or tested positive (>0.00 g/dL) with a breathalyzer. The AUDIT was administered during the registry process, by either a research assistant or research nurse depending on availability. The SMS booster procedure was explained to participants during the informed consent process. Participants were excluded if they did not speak Swahili, were too ill or unable to communicate, lacked an SMS-capable mobile phone, or if they declined informed consent.

## Booster development

In order to prolong the potential impact of the in-person PPKAY, SMS boosters were created to reinforce the knowledge/awareness of alcohol related harms, and goal-setting and self-efficacy components of the PPKAY. While further details on this multi-step process of the PPKAY and SMS development is described in full elsewhere [29, 38], in short, these short standard or personalizable phrases focused awareness, self-efficacy, and goal setting were translated, acculturated and validated through multiple rounds of translations, piloting in our research team, clinicians and the target population.

## Overview of PPKAY study design and SMS booster integration

This feasibility study followed a pilot, three-stage, single-blind, pragmatic adaptive randomized controlled study. Participants were enrolled and randomized into a usual care arm or intervention arm. Participants randomized to intervention arms underwent a 15-minute, four-step discussion (raise the subject of alcohol, provide feedback, enhance motivation, and negotiate and advise) administered by a nurse, followed by an SMS booster. Participants either received (a) PPKAY without any booster, (b) PPKAY with a personalized booster, or (c) PPKAY with standard booster. In our study, we evaluated the feasibility of our three-stage proposed trial with different sets of intervention schemes. In stage 1, patients were randomized to usual care, PPKAY without booster and PPKAY with standard booster. In stage 2, we dropped usual care and continue to randomize patients to PPKAY with standard Booster or PPKAY without a

booster. In stage 3, we compared PPKAY with standard booster to PPKAY with personalized booster.

SMS booster delivery occurred once per week, and was planned between 2-4pm on Friday afternoons. This delivery date was chosen so that participants would receive booster messages prior to weekends, where our prior data demonstrate that heavier drinking behaviors occur [23, 24, 39]. If messages were unable to be sent (due to phone being off or out of range), the messaging system automatically re-attempted to send them.

### Outcome measures

We applied an adapted TIDieR checklist to assess the feasibility of the SMS portion of the PPKAY intervention (Table 1). In addition to the TIDieR checklist items, acceptability of the SMS boosters was assessed through 6-month follow-up interviews with participants.

Acceptability of intervention was determined with interviews with short open answer questions at the completion of the study. Interviews with participants allocated to any arms were conducted at 6-month follow-up. Interview questions were designed to gauge participant acceptability of the SMS booster add-on: Did the messages serve as helpful reminders? Were the messages sent at a good time of day? Did you [participants] read the messages? Was the message content useful?

## Results

### Trial participation

41 participants were randomized into the PPKAY and one of two SMS booster arms, with 23 participants allocated to the standard booster subset and 18 allocated to the personalized booster subset (Table 2). At 6-months follow-up 90% and 81% of the allocated participants remained in the standard booster and personalized booster arms, respectively. See detailed retention data in Fig 1.

### SMS system implementation

**WHAT.** A cloud-based SMS management system was used to send both standard and personalized text messages to participants. This system accepted and immediately downloaded message replies that came from participants' mobile devices. These replies were stored in a local server at the Kilimanjaro Clinical Research Institute (KCRI). The participant ID and

**Table 1. Adapted TIDier checklist used to assess intervention feasibility.**

| Study steps | TIDier Checklist Item | Standard Description | Planned Analysis for our Study |
|---|---|---|---|
| SMS System **Implementation** | WHAT | A description of the materials needed for the intervention, including what was needed for intervention delivery. | A description of SMS system, process and data security among potential participants. |
| | TAILORING | A description of how the intervention was personalized for different intervention groups. | Description of SMS personalization process |
| SMS System **Feasibility** | WHEN and HOW MUCH | A description of the number of times the intervention was delivered and over what time period. | Analysis of delivery rate of SMS messages and comparison between intended delivery time and when the message was actually received |
| | MODIFICATIONS | A description of any changes made to the intervention throughout the course of the study. | Description of any changes to the initial protocol |
| | HOW WELL | An analysis of how well the intervention adhered to the initial plan and as assessment of fidelity. | Analysis of intervention fidelity—whether the allocated intervention was correctly received. Analysis of the patient acceptability of the intervention |

**Table 2. Demographic information stratified by booster arm type.**

| | PPKAY with Standard Booster, n = 23 | PPKAY with Personalized Booster, n = 18 |
|---|---|---|
| Age, years, M (SD) | 37 (15) | 31 (12) |
| Male (n,%) | 22 (96%) | 18 (100%) |
| Years of education, mean (SD) | 8.0 (2) | 8.7 (3) |
| **Tribe,* n (%)** | | |
| Chagga | 15 (65%) | 12 (67%) |
| Saamba/Maasai/Pare | 3 (13%) | 3 (17%) |
| Other | 4 (17%) | 3 (17%) |
| **Marital Status, n (%)** | | |
| Married | 15 (65%) | 11 (61%) |
| Single/Widowed/Separated | 8 (35%) | 7 (39%) |
| **Employment, n (%)** | | |
| Professional-2 | 3 (13%) | 3 (17%) |
| Skilled Employment-3 | 4 (17%) | 6 (33%) |
| Self-employed-4 | 5 (22%) | 8 (44%) |
| Student/Other, Farmer-0/89, 5 | 11 (48%) | 1 (6%) |
| **Monthly Income** | | |
| TSH, mean (SD) | 158,809 (114,928) | 208,765 (112,404) |
| USD, mean (SD) | 70 (51) | 92 (50) |
| **Alcohol Use Information, n (%)** | | |
| BAC positive on arrival | 2 (8%) | 0 (0%) |
| AUDIT $\geq$ 8 | 18 (78%) | 12 (67%) |
| Self-reported alcohol use prior to injury | 15 (65%) | 11 (61%) |

*Proportions may not total 100% due to rounding.
**1 USD = 2271 TSH

phone number were not included in data downloads for analysis purposes; only quality improvement staff had access to this protected information. Several security features were also enacted to ensure the privacy of participants was protected. These measures included: 1) two-factor authentication was required for research assistants to access text information, 2) alerts were sent to researchers if a user account was accessed from an unknown IP address, 3) all login sessions to the SMS system were logged and able to be accessed in real time, 4) activity logs associated with the system were stored and could be viewed for 90 days. In order for participants to receive the SMS boosters, their mobile phone had to be turned on at the time the message was sent and the receiving phone had to be in a location with cellular service. Personal information from the participants was collected by research nurses during the PPKAY intervention. For participants allocated to the personalized SMS booster arm, this information was used to create tailored booster messages.

**TAILORING.** Booster messages content development is described in detail elsewhere and is outside of the focus of this manuscript. In brief, knowledge, empowerment and goal setting focused messages with evidence based efficacy at alcohol harm reduction were adapted for local linguistic, cultural and contextual needs and prepared as Standard or Personalized messages. Fig 2 displays the work flow used to create the personalized text message boosters. Participants allocated to the personalized SMS message arm received boosters customized specifically to the person's responses during the nurse-led PPKAY. Messages were created

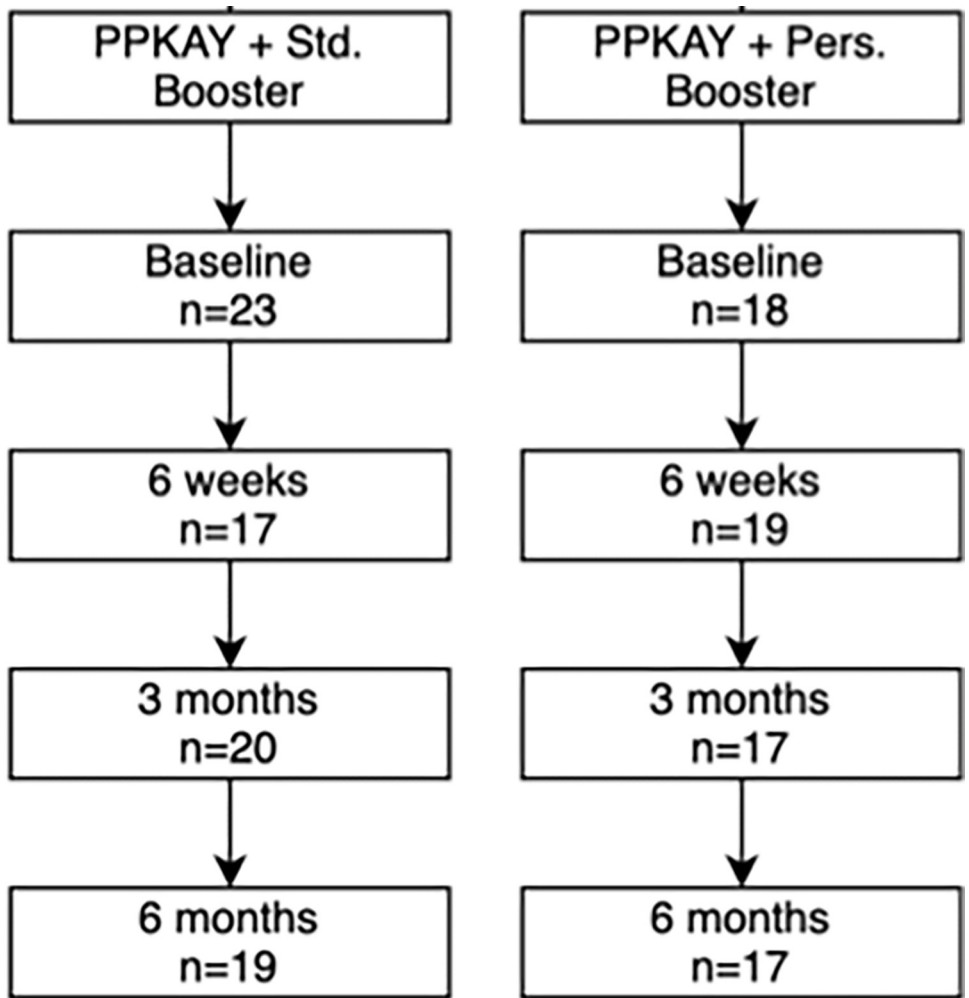

**Fig 1. Eligibility, randomization, and retention flowchart for the two booster arms.**

based on the participant's unique reasons for wanting to change their alcohol-related behavior. Such reasons were then used to fill in one of six *a priori* created outlined texts (Table 3). Quality improvement practices occurred once per week before booster messages were sent out. These practices were to ensure that the content of the personalized messages aligned with the pre-outlined templates; research staff read the Swahili text messages for language and content to align with the knowledge, empowerment and goal setting goals of the texts. Once participants were discharged from KCMC, the personalized message was sent to participants' phones in rotating fashion for the study duration.

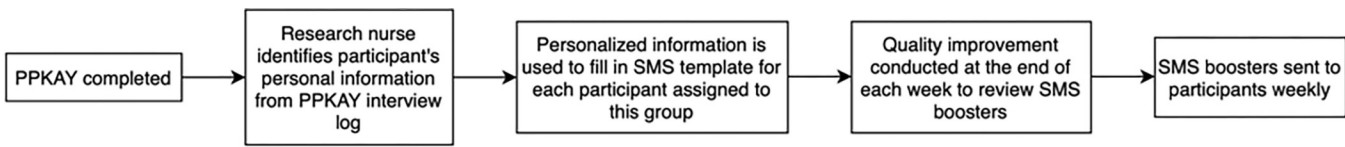

**Fig 2. Workflow diagram for the creation of personalized SMS messages.**

**Table 3. Examples of three pre-outlined SMS text messages (translated from Swahili to English) sent to participants in the personalized booster arm.** These messages were completed with personalized information collected during the PPKAY interview.

| | |
|---|---|
| Outline #1 | "Remember to try to keep your number of drinks goal below _______ for one drinking session to achieve your goal of _______." |
| Outline #2 | "Any amount of alcohol before driving a car or motorcycle has been shown to increase your risk of injury. Do not drink before driving to achieve your goal of _______." |
| Outline #3 | "To achieve your goal of _______, change your drinking habits. Even if you drink one drink or less than one drink a day." |
| Outline #4 | "Reach your goal of [. . .], avoid alcohol on weekdays." |
| Outline #5 | "Drinking too much can spoil a good night out and make you regret things. Be a good [father, mother, husband, son, daughter.] Stay safe, stay sober!" |
| Outline #6 | "Keep your goal of [xxx] and reduce your drinking this weekend. You can do it!" |

## SMS system feasibility

**WHEN and HOW MUCH.** Each of 41 study participants who were allocated to receive the intervention included receiving 12 SMS over a three-month period, for a total of 492 total messages. In total 38 received messages correctly, 1 received a message incorrectly and 3 did not receive messages. Of the 39 study participants who received messages (38 correctly, and 1 incorrectly), between both the standard and personalized booster arm, a total of 258 text messages (52%) were attempted to be sent throughout the study period. The majority of unattempted messages were not transmitted due to errors in the programming of the SMS system. Of these 258 messages, 7 were excluded from this study as they were extraneous test SMS messages that could not be traced to a specific study participant. Of the 251 remaining messages, 183 were successfully delivered. SMS delivery was intended for Friday afternoons, between 2-4pm. In practice, however, SMS boosters were registered as delivered over the course of the week (Table 4), likely due to technical/telecommunication issues (ie. cell phone being off during initial delivery attempt). 86 SMS boosters, or 34% of the total number of attempted messages, were delivered during the predetermined Friday afternoon time periods, and 15 more were delivered within 24 hours of this time period (Tables 5 and 6). The majority of messages correctly sent on Friday, within which specifically between 2-4pm, were sent through Stage 3 in the "Brief intervention + Personalized Booster" arm (Tables 5 and 6). If messages were not able to be received by the participants' phone immediately upon being sent, the messaging system then re-attempted sending the messages each day until a time out period was reached (3 days). Reasons for failed SMS delivery were often due to the participant traveling outside of cellular coverage range and that the receiving phone was turned off at the time of attempted delivery. There were 82 (33%) of the 258 messages that failed delivery. While only one study participant of the 39 who received messages received the correct 12 SMS, 6 (15.38%) received at least 10, with 2 receiving more than the 12 intended and the remaining 36 receiving less.

**Table 4. SMS delivery status by day sent.**

| Status | Monday | Tuesday | Thursday | Friday | Saturday | Sunday | Total |
|---|---|---|---|---|---|---|---|
| % delivered across all stages | 65% (11/17) | 0% (0/11) | 0% (0/0) | 76% (153/201) | 100% (15/15) | 57% (4/7) | 68% (183/271) |
| % delivered Stage 1 | 0% (0/3) | 0% (0/4) | 0% (0/0) | 76% (25/33) | 0% (0/0) | 0% (0/0) | 63% (25/40) |
| % delivered Stage 2 | 79% (11/14) | 0% (0/4) | 0% (0/0) | 79% (31/39) | 100% (11/11) | 0% (0/0) | 83% (53/64) |
| % delivered Stage 3 | 0% (0/0) | 0% (0/3) | 0% (0/0) | 75% (97/129) | 100% (4/4) | 57% (4/7) | 75% (105/140) |

**Table 5. SMS delivery status within the intended Friday 2-4pm time period, by stage.**

| Status | Stage 1 (n = 40) | Stage 2 (n = 68) | Stage 3 (n = 143) | Total (n = 251) |
|---|---|---|---|---|
| Delivered as intended (Friday between 2-4pm) | 0 (0%) | 5 (7%) | 81 (57%) | 86 (34%) |
| Delivered within 24hrs of intended time period | 0 (0%) | 11 (16%) | 4 (3%) | 15 (6%) |
| Not delivered within 24hrs of intended time period | 15 (38%) | 15 (22%) | 38 (27%) | 68 (27%) |
| Not delivered | 25 (63%) | 37 (54%) | 20 (14%) | 82 (33%) |

**Table 6. SMS delivery status within the intended Friday 2-4pm time period, by arm.**

| Status | Brief Negotiational Intervention | BNI + Personalized Booster | BNI + Standard Booster |
|---|---|---|---|
| Delivered as intended (Friday between 2-4pm) | 0 | 69 | 17 |
| Delivered within 24hrs of intended time period | 0 | 4 | 11 |
| Not delivered within 24hrs of intended time period | 0 | 18 | 50 |
| Not delivered | 4* | 18 | 60 |

*message all sent to one patient

**Table 7. Delivery of correct SMS text messages stratified by intervention arm.**

| Randomization Arm | Received Correct Intervention | Did Not Receive Correct Intervention | % Correct |
|---|---|---|---|
| Usual Care | 10 | 0 | 100% |
| PPKAY | 20 | 1 | 95% |
| PPKAY + Standard Booster | 21 | 3 | 88% |
| PPKAY+ Personalized Booster | 16 | 4 | 80% |
| TOTAL | 67 | 8 | 89% |

**MODIFICATIONS.** While no changes to the protocol were made during the course of the study timeframe, we had a continuous quality improvement process weekly to try to improve the personalized SMS based text messages as described above in TAILORING.

**HOW WELL.** *Participant allocation.* Overall, across all study groups and throughout the entire study period, 89% of participants received the correct intervention (Table 7); specifically meaning receiving the intervention they were randomized/allocated to. Rates of successful intervention provision differed between intervention groups: 100% of participants assigned to the usual care group (n = 10) received the correct intervention; 95% of participants assigned to the BNI only group (n = 21) received the correct intervention; 88% of participants allocated to the BNI + standard SMS booster group (n = 24) received the correct intervention; and 80% of participants allocated to the BNI+ personalized SMS booster group (n = 20) received the correct intervention.

**ACCEPTABILITY.** At 6-month follow-up, 33 (85%) of the 39 eligible participants (those who received SMS messages) agreed to complete follow-up interviews. Of these, 11 responded to the question about the impact of SMS boosters on their drinking behavior, with 91% (n = 10) reporting a large positive impact and 9% (n = 1) a small positive impact. 1 of the 11 (9%) received SMS messages, but was not allotted to; the remaining 10 correctly were allotted to and received messages. Of these 11, 10 (91%) reported feedback regarding satisfaction of the timing of when text messages were sent, with 9 of these (90%) noting they were satisfied with the timing due to opportunities to read the message, and 1 (10%) participant not knowing as they did not remember.

The SMS system allowed for participants to respond to the boosters with reply messages—these messages were stored in the system and recorded for analysis. Throughout the study period, 33 text messages were sent back to the SMS system by 8 unique participants. Of these participants, 1 responded with thanks for having received the message, 2 responded with direct answers to the boosters' prompts, and 3 responded with confusion as to why they were receiving the SMS boosters. Participants were given the option to unsubscribe and stop receiving text messages by replying directly to the message with either 'STOP' or 'S.' By the 6-month follow-up, 4 participants had opted to stop receiving SMS boosters.

## Discussion

This study aimed to evaluate the feasibility, acceptability, and fidelity of a SMS booster intervention that accompanied a nurse-led hospital based brief negotiational intervention for alcohol harm reduction for injury patients. Overall, this project identified an overall feasible and acceptable practice of using SMS boosters in this system, but a great need for a continuous monitoring system to ensure intervention fidelity in terms of the intensity of the SMS booster received.

### Feasibility of the SMS process

We saw acceptable rates of SMS booster delivery to participants' mobile devices (73%). This study provided support that an acceptable system is in place in this setting for SMS-focused interventions, that does not place participants at further/outside risk of private health information being shared; and that these boosters can be integrated into a research structure given the ubiquity of phones in this setting, supporting existing literature [20, 23, 40, 41]. In addition, we found that personalized text-based messages can be created in real time by study staff allowing for trialing of personalized versus standard messages, and believe our study to be one of few that utilizes personalized SMS [42] a novel intervention to the Tanzanian setting. We did notice a reduction in the correctly delivered intervention with the increasing complexity of intervention procedures (Table 7). While this supports our apriori concern that more complex interventions are more difficult to implement in low resource settings, our relatively high overall successful intervention delivery and small numbers limit our ability to provide more inferences in this regard.

### Acceptability

We found high rates of acceptability from participants who received the SMS boosters. Participants generally reported that they enjoyed receiving the messages and wanted to continue to receive them, as well as did not 'opt out' frequently. Additionally, the SMS add-on was easily implemented and did not place additional strain on the research and nursing staff conducting the PPKAY. Minimal maintenance and upkeep of the SMS system was required. While the ability and process of opting out was explained by local research nurses during the informed consent process, there were 3 participants who responded with confusion regarding the boosters; this is likely due to a common practice of sharing or changing phone numbers particularly after an injury, and as such the need for further description of the process so that this confusion or potential loss of confidentiality does not arise in future studies. While we acknowledge that phone sharing can pose a potential risk to patient health information, we performed in-depth disclosures during the informed consent process. During the consent, we disclosed possible SMSs to be sent to their phone, asking for an appropriate phone number, and emphasizing that participants can change the phone number for which boosters would be delivered at any time. In addition, the language used in the boosters, as much as possible, was positively

worded and avoided displaying potentially personal information. The changes to drinking goals could be considered personal information and at risk if shared. Although this is an important risk, in preliminary discussions, our participant population gave feedback that this was not a concern. However, this is an important point that needs to be considered when consenting participants in this kind of study, and mitigation strategies must be thought out. Our prior work has demonstrated that family members in this setting, to a great extent, are aware of family member alcohol use habits as demonstrated by convergent AUDIT scores [43]. Similarly, phone sharing potentially also allowed our participants to have more motivation to enroll family members in being a positive support or advocate for their behavior change.

### Intervention fidelity

While we found feasibility and acceptability to be high, intervention fidelity, meaning the delivery of SMS boosters was moderate but did improve during the course of the study. Our study identifies a clear need for a rigid, reactive quality improvement process that identifies in which situations SMS are not delivered to ensure that issues of fidelity we experienced do not occur in further studies. Literature has shown mixed fidelity to SMS-based alcohol interventions [44–46]. While we do not have information as to reasons for our lack of fidelity, we believe it is due to individual cell phone issues: numbers changing, being off, or getting lost. Alternative methods of locating and contacting patients, as well as methods to ensure correct phone numbers and to ensure more frequent message send-out rates or longer time-out periods, are essential for future work. Potential improved processes for ensuring fidelity could include the following: personalized message content review, frequent review of message delivery, frequent review of unsolicited incoming messages received from participants, and protocols for rectifying and documenting deviations.

### Limitations

This study was a preliminary feasibility trial. As such, this study was limited in design in terms of cohort size, scope, and ability to generalize to a larger population. We found high rates of acceptability during follow-up interviews, however, only one participant received the full and intended intervention in terms of the correct number of SMS messages delivered. Bias may be present in the in-person follow-up interviews; while we are unaware of participants reason for not partaking in the interviews, a negative experience could have been one reason biasing the results.

### Conclusion

In this trial feasibility study, successful recruitment, randomization, and allocation of participants to their assigned SMS intervention arm were assessed through a modified TIDieR checklist. This analysis has shown that SMS integration into a nurse-led BNI aimed at reducing harmful alcohol use is feasible in this Tanzanian setting. We have also shown that the SMS booster addition was acceptable and desirable by participants. During a full-scale trial to broaden the study scope and generalizability, and to determine the effectiveness of the SMS boosters would also require a rigid, reactive quality improvement process to ensure fidelity.

### Supporting information

**S1 Questionnaire. Inclusivity in global research questionnaire.**
(DOCX)

## Acknowledgments

We thank our incredibly dedicated research team at the KCMC Emergency Department for their determination and commitment to this and our other projects over the past 10 years.

## Author Contributions

**Conceptualization:** Catherine A. Staton, Blandina T. Mmbaga, Joao Ricardo Nickenig Vissoci.

**Data curation:** Ashley J. Phillips, Msafiri Pesambili, Blandina T. Mmbaga, Kennedy Ngowi.

**Formal analysis:** Mary Catherine Minnig, Arthi S. Kozhumam, Joao Ricardo Nickenig Vissoci.

**Funding acquisition:** Catherine A. Staton, Blandina T. Mmbaga.

**Methodology:** Mary Catherine Minnig, Arthi S. Kozhumam, Joao Ricardo Nickenig Vissoci.

**Project administration:** Ashley J. Phillips, Msafiri Pesambili.

**Supervision:** Catherine A. Staton, Blandina T. Mmbaga, Kennedy Ngowi.

**Writing – original draft:** Mary Catherine Minnig, Arthi S. Kozhumam.

**Writing – review & editing:** Catherine A. Staton, Ashley J. Phillips, Msafiri Pesambili, Brian Suffoletto, Blandina T. Mmbaga, Kennedy Ngowi, Joao Ricardo Nickenig Vissoci.

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
