## [Decision Letter · Decision Letter 0]

18 Feb 2022

PGPH-D-21-01151

Feasibility of SMS Booster for Alcohol Reduction in Injury Patients in Tanzania

Dear Dr. Staton,

Thank you for submitting your manuscript to PLOS Global Public Health. After careful consideration, we feel that it has merit but does not fully meet PLOS Global Public Health’s publication criteria as it currently stands. Therefore, we invite you to submit a revised version of the manuscript that addresses the points raised during the review process.

We look forward to receiving your revised manuscript.

Kind regards,

Joel Msafiri Francis, MD, MS, PhD

Academic Editor

Journal Requirements:

1. Please include a complete copy of PLOS’ questionnaire on inclusivity in global research in your revised manuscript. Our policy for research in this area aims to improve transparency in the reporting of research performed outside of researchers’ own country or community. The policy applies to researchers who have travelled to a different country to conduct research, research with Indigenous populations or their lands, and research on cultural artefacts. The questionnaire can also be requested at the journal’s discretion for any other submissions, even if these conditions are not met.  Please find more information on the policy and a link to download a blank copy of the questionnaire here: https://journals.plos.org/globalpublichealth/s/best-practices-in-research-reporting. Please upload a completed version of your questionnaire as Supporting Information when you resubmit your manuscript.”

2. Please provide separate figure files in .tif or .eps format only.  Please ensure that all files are under our size limit of 20MB.  

For more information about how to convert your figure files please see our guidelines: Once you've converted your files to .tif or .eps, please also make sure that your figures meet our format requirements

3. Since your data is not available for proprietary reasons, please explain via email why the data is not available. Please also include the contact information for the third party organization that should be contacted should other researchers want to request access to this data and please include the full citation of where the data can be found. We also request that you verify with us via email that any researcher will be able to obtain the data set in the same manner that the you have obtained it. If you feel you are unwilling or unable to adhere to this policy, please explain your reasons by return email and your exemption request will be escalated to the editor for approval. Your exemption request will be handled independently and will not hold up the peer review process, but will need to be resolved should your manuscript be accepted for publication. One of the Editorial team will be in touch if they require more information.

4. Please amend your detailed Financial Disclosure statement. This is published with the article, therefore should be completed in full sentences and contain the exact wording you wish to be published.

ii). State the initials, alongside each funding source, of each author to receive each grant.

iii). State what role the funders took in the study. If the funders had no role in your study, please state: “The funders had no role in study design, data collection and analysis, decision to publish, or preparation of the manuscript.”

Additional Editor Comments (if provided):

Reviewers' comments:

Reviewer's Responses to Questions

**Comments to the Author**

1. Does this manuscript meet PLOS Global Public Health’s publication criteria? Is the manuscript technically sound, and do the data support the conclusions? The manuscript must describe methodologically and ethically rigorous research with conclusions that are appropriately drawn based on the data presented.

Reviewer #1: Yes

Reviewer #2: Yes

2. Has the statistical analysis been performed appropriately and rigorously?

Reviewer #1: Yes

Reviewer #2: Yes

3. Have the authors made all data underlying the findings in their manuscript fully available (please refer to the Data Availability Statement at the start of the manuscript PDF file)?

Reviewer #1: No

Reviewer #2: Yes

4. Is the manuscript presented in an intelligible fashion and written in standard English?

Reviewer #1: Yes

Reviewer #2: Yes

5. Review Comments to the Author

Reviewer #1: This is an interesting manuscript describing a feasibility study of a brief intervention supplemented by SMS boosters, with the aim of reducing alcohol use among persons attending the injury ward with harmful and hazardous alcohol use at admission, in Moshi, Tanzania. The study addresses a very important problem, targets a potentially teachable population, and includes leveraging the power of SMS to potentially increase the efficacy of the in-person intervention. Overall, the methods are sound and the writing is clear. However, the complex study design is a bit confusing and detracts from the presentation, and there are some missing pieces of information which detract from the conclusions. Some specific comments follow.

1. Overall: the presentation of the results using the checklist framework is novel, however it makes the paper hard to read. It seems that the results are a combination of intervention development and feasibility combined. I think perhaps the whole paper would be easier to read if these were clearly separated. Feasibility seems most relevant after the intervention has been completely developed, i.e. the kinks of sending the messages etc. are worked out.

2. Line 130 – is there a reference for the description of the local adaptation?

3. Line 166 – it is not described when and by whom the AUDIT was administered. Also the measure of self-reported alcohol use prior to injury is reported but methods of ascertainment not described.

4. Lines 172-186 and Figure 1. The adaptive RCT design is confusing. It is not clear why different arms are dropped at different stages, except for the reason given for Stage 3. Either more or less detail is needed. For more detail, I wondered how the stages occurred over time, and again, what the decision points were. Figure 1 does not depict the design, just the final allocation, which somehow made me feel like the authors felt the adaptive piece was not relevant which would argue for simplifying the text.

5. There is no description of the BNI and boosters. These do not need to be lengthy, but do need to be included.

6. It is confusing that the paper says its focus is on the SMS portion of the intervention, after describing the complex study design. Results for the entire group are given throughout the paper.

7. There is high completion rate so Table 2b is not needed.

8. Lines 242-244. The paper states that phones had to be on and in cell service range to receive messages, but my understanding is that messages will be sent later if a phone is off or out of range. This is suggested by line 258 and lines 264-266 and should be clarified in the methods.

9. Were there any safeguards in place to ensure that the intended recipient got the message, rather than others with whom phones may be shared? PINs have been recommended.

10. Lines 249-255 The paper states that 492 total messages were intended, but only 258 were attempted. There is no discussion of the 234 messages not attempted – why weren’t they attempted?

11. Lines 271-272 and Table 3 and Table 4a. The observation that SMS delivery during the phases of the study is not helpful given that the reader is not told what measures were taken across phases, and what the phases really were exactly, other than changes in allocations to study arms.

12. Line 288 and Table. 5. Six reasons are mentioned but only 3 given in the table. Give all 6.

13. Lines 324-327 are repeats of lines 198-200.

14. Lines 330-337. It says 33 people agreed to answer questions but only 11 responded? Why didn’t the other 22 answer the question? Wouldn’t the non-response impact interpretation of the results?

15. Line 350 – says nurse-led BNI – really not described prior. Should be in the methods.

16. Line 357 – 73% success rate for SMS – not sure where this number comes from as the results are hard to read.

17. Line 358 – “secure system is in place”. I didn’t see this described elsewhere.

Reviewer #2: This paper looked at the feasibility of using SMS messaging as part of an alcohol harm reduction intervention following injury in Tanzania. This is an interesting paper on a very important topic and the feasibility of the intervention was well described within TIDier checklist framework

Minor comments:

Due to the Small n in some places there are very small numbers in each cell of the table – where there are less than 3 participants per cell suggest suppressing data to limit potential identification of participants

Table 4- these tables were hard to follow in terms of referring to stage and phase of the study – it would be worth reviewing these and revising the labelling

Table 6 – it was not clear why some participants did not receive the intervention they were assigned to.

6. PLOS authors have the option to publish the peer review history of their article (what does this mean?). If published, this will include your full peer review and any attached files.

**Do you want your identity to be public for this peer review?** For information about this choice, including consent withdrawal, please see our Privacy Policy.

Reviewer #1: No

Reviewer #2: No

---

## [Decision Letter · Decision Letter 1]

27 Jun 2022

PGPH-D-21-01151R1

Feasibility of SMS Booster for Alcohol Reduction in Injury Patients in Tanzania

Dear Dr. Staton,

Thank you for submitting your manuscript to PLOS Global Public Health. After careful consideration, we feel that it has merit but does not fully meet PLOS Global Public Health’s publication criteria as it currently stands. Therefore, we invite you to submit a revised version of the manuscript that addresses the points raised during the review process.

This remains an interesting  and well-presented paper. However the resubmission still needs some minor revisions, as requested by the Reviewers below. Please address those points and re-submit to the system.

We look forward to receiving your revised manuscript.

Kind regards,

Dr Isabelle Uny

Academic Editor

Journal Requirements:

1. Please amend your online detailed Financial Disclosure statement. This is published with the article, therefore should be completed in full sentences and contain the exact wording you wish to be published.

State what role the funders took in the study. If the funders had no role in your study, please state: “The funders had no role in study design, data collection and analysis, decision to publish, or preparation of the manuscript.”

2. Please update your online Competing Interests statement. If you have no competing interests to declare, please state: “The authors have declared that no competing interests exist.”

Additional Editor Comments (if provided):

Reviewers' comments:

Reviewer's Responses to Questions

**Comments to the Author**

1. If the authors have adequately addressed your comments raised in a previous round of review and you feel that this manuscript is now acceptable for publication, you may indicate that here to bypass the “Comments to the Author” section, enter your conflict of interest statement in the “Confidential to Editor” section, and submit your "Accept" recommendation.

Reviewer #1: (No Response)

Reviewer #3: (No Response)

2. Does this manuscript meet PLOS Global Public Health’s publication criteria? Is the manuscript technically sound, and do the data support the conclusions? The manuscript must describe methodologically and ethically rigorous research with conclusions that are appropriately drawn based on the data presented.

Reviewer #1: Yes

Reviewer #3: Yes

3. Has the statistical analysis been performed appropriately and rigorously?

Reviewer #1: Yes

Reviewer #3: Yes

4. Have the authors made all data underlying the findings in their manuscript fully available (please refer to the Data Availability Statement at the start of the manuscript PDF file)?

Reviewer #1: No

Reviewer #3: Yes

5. Is the manuscript presented in an intelligible fashion and written in standard English?

Reviewer #1: Yes

Reviewer #3: Yes

6. Review Comments to the Author

Reviewer #1: This continues to be an interesting manuscript. The authors have addressed my prior comments but there are a few remaining issues.

1. Line 52 – include “negotiated” in the spelling out of BNI

2. Lines 65 and 66 and elsewhere: The term “supposed to” sounds unscientific. Replace with crisper terminology?

3. Line 71: include the word “interviewed”, since the majority were not interviewed

4. Line 172: I believe accentuate is meant instead of attenuate

5. Lines 241-249. How were the messages developed?

6. Tables 3, 4a, 4b: Since the stages (or phases) are no longer described (which was helpful), the phases and stages listed in these tables are hard to interpret. I know the point is that the delivery improved throughout the study, but maybe this can be said more simply (the extra detail in the tables is not really needed).

7. Table 4a: Too many digits in the %s given

8. Line 297: describe what the quality improvement entailed

9. Lines 306-309 and Table 6: Since the focus changed to only the SMS part of the study, data on the other arms are not needed.

10. Line 331: Drop “only”, it is editorializing

11. Lines 344-346: I do think drinking goals is private health information, so I would not assert no info could be shared.

12. Lines 362-364: Not a sentence.

13. Lines 372-373: I don’t agree with this statement, see #11.

14. Lines 373-377: Are you suggesting that family members intercepted the messages? It sounds odd and not related to data presented in the MS.

15. Lines 395-397: I think you are talking about the missing 22 interviews. Be explicit that you don’t know why they were not completed and that could have caused bias.

Reviewer #3: 1. In the opening sentence, suggested to replace ‘is the leading cause of death’ with ‘contributes to health conditions that cause more than 60% of deaths.

2. To a reader who is not familiar with the BNI approach, more details need to be provided on how it works and why was it the most appropriate technique for this particular study.

3. The BNI may not be a reliable technique for arriving at a conclusion regarding the feasibility of an intervention that takes longer (let say, than a month, two, or three). It becomes even complex when the patient has to recall the changes that have taken place over an extended period of time.

4. In the TIDieR framework, the ‘WHAT’ item is limited to ‘materials needed’ to deliver the intervention. What about the actual content of the SMSs? (consider how it affects responses).

5. In the case of acceptability, if there was no a continuous monitoring of patients’ response as soon as they received the messages, how does one establish (if not informed by a feedback over a prolonged time) that they (participants) ‘enjoyed’ the messages?

7. In the findings, why is fidelity limited to receiving the messages. Were all planned activities implemented as planned? Consider the timing, techniques, etc. In most cases, on the side of the study and study team and not just on the patients. Was there a delivery confirmation strategy? Why didn’t it work? In well-planned interventions, reasons for deviating from plans and milestones are important. Were there any monitoring, evaluation, and learning plans and strategies?

8. There is a contradiction between the information provided in the abstract, methods, findings, and limitations’ sections regarding the number of participants who received a full and intended intervention package in terms of SMSs. Is the correct number 38 or 1?

9. The conclusion needs to provide key ‘take-home’ lessons regarding the feasibility, the forces that affect the intervention in both positive and negative ways.

7. PLOS authors have the option to publish the peer review history of their article (what does this mean?). If published, this will include your full peer review and any attached files.

**Do you want your identity to be public for this peer review?** For information about this choice, including consent withdrawal, please see our Privacy Policy.

Reviewer #1: No

Reviewer #3: **Yes: **Respicius Shumbusho Damian

---

## [Decision Letter · Decision Letter 2]

12 Sep 2022

PGPH-D-21-01151R2

Feasibility of SMS Booster for Alcohol Reduction in Injury Patients in Tanzania

Dear Dr. Staton,

Thank you for submitting your manuscript to PLOS Global Public Health. After careful consideration, we feel that it has merit but does not fully meet PLOS Global Public Health’s publication criteria as it currently stands. Therefore, we invite you to submit a revised version of the manuscript that addresses the points raised during the review process.

We look forward to receiving your revised manuscript.

Kind regards,

Isabelle Uny

Academic Editor

Journal Requirements:

Additional Editor Comments (if provided):

Please look at the Minor revisions asked by Reviewer 1 on your second revision and re-submit.

Thanks

Reviewers' comments:

Reviewer's Responses to Questions

**Comments to the Author**

1. If the authors have adequately addressed your comments raised in a previous round of review and you feel that this manuscript is now acceptable for publication, you may indicate that here to bypass the “Comments to the Author” section, enter your conflict of interest statement in the “Confidential to Editor” section, and submit your "Accept" recommendation.

Reviewer #1: (No Response)

Reviewer #3: All comments have been addressed

2. Does this manuscript meet PLOS Global Public Health’s publication criteria? Is the manuscript technically sound, and do the data support the conclusions? The manuscript must describe methodologically and ethically rigorous research with conclusions that are appropriately drawn based on the data presented.

Reviewer #1: Yes

Reviewer #3: Yes

3. Has the statistical analysis been performed appropriately and rigorously?

Reviewer #1: Yes

Reviewer #3: Yes

4. Have the authors made all data underlying the findings in their manuscript fully available (please refer to the Data Availability Statement at the start of the manuscript PDF file)?

Reviewer #1: No

Reviewer #3: Yes

5. Is the manuscript presented in an intelligible fashion and written in standard English?

Reviewer #1: Yes

Reviewer #3: Yes

6. Review Comments to the Author

Reviewer #1: • BNI is spelled out inconsistently between the abstract and the main text.

• Lines 99-103 – the word generally appears 3 times.

• Table 5 comes before Tables 3 and 4.

• Table 3 retains the stages but these are not described in the text. Please describe.

• R1. Comment 11. The authors give a lengthy answer about their findings about the participants’ feelings about the risk of sharing private information in their response to the reviewers but I do not see that discussion included in the manuscript. Please include this discussion in the manuscript.

• In future responses to reviewers, please include the changes to the text within the response letter, to make it easier for the reviewer to see how the manuscript was edited without having to go hunting in the manuscript. I found it difficult to see what changes were made.

Reviewer #3: The authors have addressed the comments and responded to the queries raised about both the subject and the study approach. At this point, they can do the final checks to eliminate the minor errors relating to the choice of terms and structure construction of statements.

The only are that needs attention is the assumption in the background that harmful alcohol use is more prevalent in developing or LMICs such as Tanzania. Experiences show that there are well-resourced countries such as Russia in which alcohol use is higher than it is in Tanzania and many other LMICs. Therefore, as also pointed out in the background, emphasis should be placed on the contextualized definition of the health hazards caused by alcohol use in Tanzania.

The perception among both authorities and health workers, especially due to majority poverty and the pressure placed by the burden of other diseases on the health systems, is that alcoholism and alcohol related health hazards is a 'non-issue'. Therefore, both preventive and curative measures are not a priority of the healthcare system. This would capture the reality on the ground.

7. PLOS authors have the option to publish the peer review history of their article (what does this mean?). If published, this will include your full peer review and any attached files.

**Do you want your identity to be public for this peer review?** For information about this choice, including consent withdrawal, please see our Privacy Policy.

Reviewer #1: No

Reviewer #3: **Yes: **Dr. Respicius Shumbusho Damian

---

## [Decision Letter · Decision Letter 3]

3 Nov 2022

Feasibility of SMS Booster for Alcohol Reduction in Injury Patients in Tanzania

PGPH-D-21-01151R3

Dear Dr. Staton,

We are pleased to inform you that your manuscript 'Feasibility of SMS Booster for Alcohol Reduction in Injury Patients in Tanzania' has been provisionally accepted for publication in PLOS Global Public Health.

Best regards,

Joseph El-Khoury, MD MSc FRCPsych

Academic Editor

Reviewer Comments (if any, and for reference):

Reviewer's Responses to Questions

**Comments to the Author**

1. If the authors have adequately addressed your comments raised in a previous round of review and you feel that this manuscript is now acceptable for publication, you may indicate that here to bypass the “Comments to the Author” section, enter your conflict of interest statement in the “Confidential to Editor” section, and submit your "Accept" recommendation.

Reviewer #1: All comments have been addressed

Reviewer #3: All comments have been addressed

2. Does this manuscript meet PLOS Global Public Health’s publication criteria? Is the manuscript technically sound, and do the data support the conclusions? The manuscript must describe methodologically and ethically rigorous research with conclusions that are appropriately drawn based on the data presented.

Reviewer #1: Yes

Reviewer #3: Yes

3. Has the statistical analysis been performed appropriately and rigorously?

Reviewer #1: Yes

Reviewer #3: Yes

4. Have the authors made all data underlying the findings in their manuscript fully available (please refer to the Data Availability Statement at the start of the manuscript PDF file)?

Reviewer #1: Yes

Reviewer #3: No

5. Is the manuscript presented in an intelligible fashion and written in standard English?

Reviewer #1: Yes

Reviewer #3: Yes

6. Review Comments to the Author

Reviewer #1: All good, thanks for the revision.

Reviewer #3: The authors have addressed most of my concerns and queries. At this point I leave the decision to the editor

7. PLOS authors have the option to publish the peer review history of their article (what does this mean?). If published, this will include your full peer review and any attached files.

**Do you want your identity to be public for this peer review?** For information about this choice, including consent withdrawal, please see our Privacy Policy.

Reviewer #1: No

Reviewer #3: **Yes: **Dr. Respicius Shumbusho Damian,
